# Delta-Radiomics Predicts Response to First-Line Oxaliplatin-Based Chemotherapy in Colorectal Cancer Patients with Liver Metastases

**DOI:** 10.3390/cancers14010241

**Published:** 2022-01-04

**Authors:** Valentina Giannini, Laura Pusceddu, Arianna Defeudis, Giulia Nicoletti, Giovanni Cappello, Simone Mazzetti, Andrea Sartore-Bianchi, Salvatore Siena, Angelo Vanzulli, Francesco Rizzetto, Elisabetta Fenocchio, Luca Lazzari, Alberto Bardelli, Silvia Marsoni, Daniele Regge

**Affiliations:** 1Department of Surgical Sciences, University of Turin, 10124 Turin, Italy; arianna.defeudis@unito.it (A.D.); giulia.nicoletti@unito.it (G.N.); simone.mazzetti@ircc.it (S.M.); daniele.regge@ircc.it (D.R.); 2Radiology Unit, Candiolo Cancer Institute, FPO-IRCCS, 10060 Candiolo, Italy; laura.pscd@gmail.com (L.P.); giovanni.cappello@ircc.it (G.C.); 3Department of Oncology and Hemato-Oncology, Università degli Studi di Milano, 20122 Milan, Italy; andrea.sartorebianchi@unimi.it (A.S.-B.); salvatore.siena@unimi.it (S.S.); angelo.vanzulli@unimi.it (A.V.); 4Niguarda Cancer Center, Grande Ospedale Metropolitano Niguarda, 20162 Milan, Italy; 5Department of Radiology, ASST Grande Ospedale Metropolitano Niguarda, 20162 Milan, Italy; francesco.rizzetto@unimi.it; 6Multidisciplinary Outpatient Oncology Clinic, Candiolo Cancer Institute, FPO-IRCCS, 10060 Candiolo, Italy; elisabetta.fenocchio@ircc.it; 7Precision Oncology, IFOM-The FIRC Institute of Molecular Oncology, 20139 Milan, Italy; luca.lazzari@ifom.eu (L.L.); silvia.marsoni@ifom.eu (S.M.); 8Candiolo Cancer Institute-FPO, IRCCS, 10060 Candiolo, Italy; alberto.bardelli@unito.it; 9Department of Oncology, University of Torino, 10060 Candiolo, Italy

**Keywords:** delta-radiomics, prediction, machine learning, CRC liver metastases, response to therapy, artificial intelligence

## Abstract

**Simple Summary:**

Oxaliplatin-based chemotherapy remains the mainstay of first-line therapy in patients with metastatic colorectal cancer (mCRC). Unfortunately, only approximately 60% of treated patients achieve response, and half of responders will experience an early onset of disease progression. Furthermore, some individuals will develop a mixed response due to the emergence of resistant tumor subclones. The ability to predicting which patients will acquire resistance could help them avoid the unnecessary toxicity of oxaliplatin therapies. Furthermore, sorting out lesions that do not respond, in the context of an overall good response, could trigger further investigation into their mutational landscape, providing mechanistic insight towards the planning of a more comprehensive treatment. In this study, we validated a delta-radiomics signature capable of predicting response to oxaliplatin-based first-line treatment of individual liver colorectal cancer metastases. Findings could pave the way to a more personalized treatment of patients with mCRC.

**Abstract:**

The purpose of this paper is to develop and validate a delta-radiomics score to predict the response of individual colorectal cancer liver metastases (lmCRC) to first-line FOLFOX chemotherapy. Three hundred one lmCRC were manually segmented on both CT performed at baseline and after the first cycle of first-line FOLFOX, and 107 radiomics features were computed by subtracting textural features of CT at baseline from those at timepoint 1 (TP1). LmCRC were classified as nonresponders (R−) if they showed progression of disease (PD), according to RECIST1.1, before 8 months, and as responders (R+), otherwise. After feature selection, we developed a decision tree statistical model trained using all lmCRC coming from one hospital. The final output was a delta-radiomics signature subsequently validated on an external dataset. Sensitivity, specificity, positive (PPV), and negative (NPV) predictive values in correctly classifying individual lesions were assessed on both datasets. Per-lesion sensitivity, specificity, PPV, and NPV were 99%, 94%, 95%, 99%, 85%, 92%, 90%, and 87%, respectively, in the training and validation datasets. The delta-radiomics signature was able to reliably predict R− lmCRC, which were wrongly classified by lesion RECIST as R+ at TP1, (93%, averaging training and validation set, versus 67% of RECIST). The delta-radiomics signature developed in this study can reliably predict the response of individual lmCRC to oxaliplatin-based chemotherapy. Lesions forecasted as poor or nonresponders by the signature could be further investigated, potentially paving the way to lesion-specific therapies.

## 1. Introduction

Colorectal cancer (CRC) is the second most frequent cancer in developed countries, and the third cause of cancer-related death in the USA and in Europe [1]. Approximately 20% of all subjects with CRC have distant-stage disease at presentation, the liver being the most frequent target of metastatic spread [2]. If not treated, patients with liver metastases exhibit an unfavorable prognosis with median overall survival (OS) shorter than 20 months [3], while hepatic resection is associated with 5-year survival rates ranging from 16% to 71% [3].

Patients with liver-only unresectable colorectal cancer (lmCRC) can achieve downsizing of their metastases by neoadjuvant treatment strategies making the lesions resectable. Approximately one third and one fourth of these patients are alive at the 5- and 10-year landmarks, which is a slightly less than that of patents with initially resectable lmCRC, but still significantly higher than patients not amenable to surgery [3]. In this setting, patients undergo first-line treatment based on the combination of 5-fluorouracil, leucovorin, and oxaliplatin (FOLFOX) or irinotecan (FOLFIRI/FOLFOXIRI) with the association of antivascular endothelial growth factor (anti-VEGF) agents or anti-epidermal growth factor receptor monoclonal antibodies (anti-EGFR), e.g., bevacizumab or cetuximab/panitumumab [4]. Unfortunately, only 60% of patients achieve partial or complete response with first-line therapy. Moreover, approximately 50% of responders will experience early onset of disease progression [5,6,7,8]. To date, besides exclusion of RAS/BRAF mutated tumors to anti-EGFR treatment, and assessment of BRAF mutations, HER2 amplification, or MSI status for targeted options, no reliable biomarkers capable of determining which patients will benefit from chemotherapy are available [9,10].

Adding to the complexity, some patients may exhibit a heterogeneous response, with some lesions shrinking and others progressing because of the expansion of resistant tumor subclones [11,12,13,14]. Being capable of distinguishing which CRC liver metastases (lmCRC) will have a lasting response, which will progress, and which will linger and eventually progress, could pave the way to truly personalized treatment [12]. For example, lesions that are predicted to be poor or nonresponders could be investigated with a biopsy to detect new dominant molecular clones, which could lead to a change in therapy, ensuring a longer period of disease control [15].

In the last few years, radiomics biomarkers derived from medical images have been proposed to predict response to therapy in several clinical contexts. However, the results of clinical trials on the prediction of response to therapy of lmCRC have yielded suboptimal results, with AUC values ranging between 0.68 and 0.79 in the validation cohorts [16,17,18].

Different factors, such as segmentation modality, CT instrumentation and protocols, and reconstruction algorithms have been shown to limit the reproducibility and robustness of radiomics features [19,20]. We hypothesize that some limits of conventional radiomics models might be overcome by delta-radiomics, the analysis of variations of the radiomics parameters extracted from CT scans performed at different timepoints. The method seems to be less influenced by instrumentation and acquisition parameters and could yield more robust and reproducible results across centers [21]. To our knowledge, only two studies have used a delta-radiomics approach in patients with lmCRC and they achieved a high accuracy in predicting overall survival (OS) [22,23].

The aim of this study was to assess if a delta-radiomics score, obtained by subtracting radiomics features of CT performed at baseline from those at timepoint one (TP1), can predict the response of individual lmCRC to first-line FOLFOX chemotherapy.

## 2. Materials and Methods

### 2.1. Study Design and Patients

This retrospective study includes lmCRC of chemo-naïve patients with stage IV CRC, treated with a FOLFOX-based first-line chemotherapy from February 2012 to June 2020. All relevant clinical and radiological data were retreated from the AlfaΩmega (AΩ-R) longitudinal observational platform, a repository which includes clinical and imaging data, aimed at empowering the development and validation of new integrated prognostic markers for CRC (NCT04120935, ClinicalTrials.gov). The study drew data from real-world patients referred to two comprehensive cancer centers in northern Italy (center A: Niguarda Cancer Center, Grande Ospedale Metropolitano Niguarda; center B: Candiolo Cancer Institute). The study was performed in accordance with the principles of the Declaration of Helsinki and the International Conference on Harmonization and Good Clinical Practice guidelines. The ethical committees from both centers approved the trial. Informed consent was waived for patients that were either nontraceable or dead at the time of recruitment (Provision no. 146 of 5 June 2019 containing the requirements relating to the processing of special categories of data, Italian Data Protection Authority).

Inclusion criteria were the availability of: (1) portal-phase CT scans performed at baseline and after the fourth cycle of first-line chemotherapy (TP1); (2) at least one lmCRC with a long-axis diameter of 10 mm or more at baseline CT; (3) contrast-enhanced CT scans up to progressive disease (PD). Exclusion criteria were: (1) poor quality CT scans, such as slice thickness > 3 mm, absence of portal-phase scan, insufficient parenchymal enhancement (measured as liver enhancement in the portal phase with an increase of liver density inferior to 30 HU from unenhanced scan); (2) patients with metachronous lmCRC who had previously undergone adjuvant chemotherapy; (3) lesions with partial response (PR) or stable disease (SD) that underwent first-line therapy for less than 8 months (for the rationale, see Section 2.3); (4) lesions that reached a complete response (CR) (i.e., that were not visible) at TP1, since radiomics features could not be extracted. Of note, in this study, maintenance therapy was considered part of the first-line treatment even if performed with drugs different from the previous ones, an approach called “switch maintenance”.

After retrieving CT examinations, data were anonymized in each center, assigning a random unique identifier to each patient. This process ensured patient privacy protection according to the most recent European data regulation policies, since no connection between the original patient information and CT-anonymized data was available. All patients from center A were used for model construction (training set), while patients from center B were adopted to externally validate the model (validation set). 

### 2.2. CT Equipment and Protocols

Exams were obtained from different CT scanners (Somatome Definition Flash—Siemens, Sensation 64—Siemens, Brilliance 64—Philips). All patients underwent contrast-enhanced CT using different contrast medium (1.5–2 mL/kg of a nonionic contrast medium [370–400 mg/mL]) injected at a rate of 3–4 mL/s, using an automatic power injector. Radiomics analysis was performed on the portal-phase acquisition, with a delay of 60–70 s after the beginning of the contrast agent injection. CT scanning parameters were the following: X-ray tube voltage: 100–140 kV; current intensity: 180–230 mA; slice thickness ≤ 3 mm; field of view 300–350 mm; 512 × 512 matrix.

### 2.3. Reference Standard

In this study, the reference standard was RECIST version 1.1 [24]. However, RECIST was assessed separately for each segmented lmCRC and not on a per-patient basis as in clinical trials. Consequently, CR was defined as the disappearance of the selected lmCRC; PR as a reduction of at least 30% of the lesion long-axis diameter; progressive disease (PD) as a lesion long-axis diameter increases of at least 20%; and SD if the lesion did not shrink enough to qualify for PR or increase enough to qualify for PD.

Based on the median progression-free survival values reported from the most relevant phase III clinical trials on first-line chemotherapy in metastatic CRC [5,6,7], lesions were dichotomized into two groups: nonresponders (R−) and good responders (R+). All lmCRC that showed PD before 8 months were considered as R−. Conversely, all lesions that either reached a CR, showed PR, or were stable for at least 8 months were considered R+. Following the RECIST 1.1 guidelines, all lesions measured at baseline had their actual measurements recorded at each subsequent TP. If the lesion had likely disappeared, the measurement was recorded as 0 mm, while if the lesion was believed to be present and was faintly seen and too small to measure, a default value of 5 mm was assigned. 

### 2.4. Features Extraction

Two resident radiologists with more than 3 years of clinical experience in abdominal imaging manually contoured a maximum of ten lmCRC with a diameter 10 mm or more per patient using ITK-Snap (http://www.itksnap.org, accessed on 28 December 2021). Each tumor volume was segmented on each slice on both the baseline and TP1 CT scans. Lesions were labeled with the same numerical code at the two timepoints. Segmentation masks were then reviewed and, if necessary, modified by an experienced radiologist (>20 years of experience in reporting CT scans). Approximately 50% of baseline segmentations were slightly revised, while 5% of them were strongly revised. Conversely, at TP1, 10% of segmentations were slightly revised.

Before extracting features, all lmCRC masks were resegmented between the 1st and the 99th percentile of the ROI to remove outliers and to make calculation of texture features tractable [25]. Additionally, all images were discretized using a fixed number of bins (*n* = 32) to introduce a normalizing effect which may be beneficial when intensity units are arbitrary or differ among centers, and where image contrast is considered important [26]. For all lmCRC, we computed the following 107 features: (1) 14 shape-based; (2) 18 intensity-based statistics, i.e., mean, 25th, 50th, 75th percentiles, skewness, kurtosis, intensity kurtosis, and intensity variance; (3) 24 features derived from the gray level co-occurrence matrices (GLCM); (4) 16 features derived from the gray level run length matrices (GLRLM); (5) 16 features derived from the gray level size zone matrix (GLSZM); (6) 5 features derived from the neighboring gray tone difference matrix (NGTDM); (7) 14 features derived from the gray level dependence matrix (GLDM). All features were computed symmetrically for each of the 4 directions of a 2D image, and then averaged. 

Features were extracted from both baseline ($Feat_{Baseline}$) and TP1 ($Feat_{TP1}$) CT scans. Then, delta-radiomics features were computed as follows:
(1)$$DeltaRadiomics=Feat_{TP1}-Feat_{Baseline}$$

Texture features were computed using PyRadiomics [27] that was compliant to the image biomarker standardization initiative (IBSI) [28]. A list of all features and parameters is provided in Appendix A.

### 2.5. Feature Selection and Radiomics Model Development

Feature selection (FS) is the process of choosing a subset of original features to reduce dimensionality, remove irrelevant data, increase learning accuracy, and improve result comprehensibility [29,30]. In this study, FS was performed using univariate analysis. First, we normalized all features using the min–max scaling method to obtain the same range of values for each feature. Then, we calculated (1) the nonparametric Mann–Whitney U test between the R+ and R− classes for each feature and excluded the nonstatistically significant (*p*-value > 0.05) parameters and (2) the area under the curve (AUC) of the remaining features was then computed to compare each feature for the differentiation of the two classes. Then we excluded all features with an AUC value statistically lower than a conventional value of 0.60. This value was chosen to consider only features that individually perform statistically better than a random classifier, i.e., AUC = 0.5.

Once the best features subset was created, different machine learning techniques were developed, including a support vector machine classifier, a logistic regression with different statistical distributions, a decision tree (DT), and a random forest. All these methods were trained using patients from center A and tested through a cross-validation. The best classifier was chosen based on the accuracy on the test set and then externally validated on patients from center B. During the validation phase, no interplatform retraining was performed. Therefore, both the cutoffs for the individual parameters and for the delta-radiomics score were maintained on the validation cohort. The outcome of all classifiers, called radiomics score, was computed as the posterior probability that the output will respond to therapy (0 = probability that the lmCRC will respond equal to 0%; 1 = 100% probability of response).

### 2.6. Statistical Analysis

Sensitivity (SE), specificity (SP), negative predictive value (NPV), and positive predictive value (PPV) of the radiomics score were computed on both the training and the validation sets. SE was defined as the ratio between the number of correctly classified R+ lmCRC over the total number of R+ lmCRC, SP as the ratio between the number of correctly classified R− lmCRC over the total number of R− lmCRC, PPV as the ratio between the number of correctly classified R+ lmCRC over the total number of lmCRC classified as R+, and NPV as the ratio between the number of correctly classified R− lmCRC over the total number of lmCRC classified as R−. The ROC curve that describes the ratio between sensitivity and specificity at different cutoffs was performed and the area under the curve (AUC) was calculated. The best cutoff was chosen as the one that maximizes the NPV on the training set in order to increase the probability of correctly identifying R− lesions that can benefit either from a pathological assessment or from interventional radiology ablative procedures. The selected cutoff was then applied on the validation set.

SE, SP, NPV, and PPV were also computed for the response based on the RECIST criteria at TP1 (lesion RECIST). For this analysis, lesions predicted R+ were those that showed PR or were SD at TP1, and R− those that showed PD at TP1. Results between the radiomics score on the validation set and the lesion RECIST were compared using the “N-1” chi-squared test, while Mann–Whitney test was used to compare the lesions’ diameters. A-value < 0.05 was considered as statistically significant.

## 3. Results

The study flowchart is shown in Figure 1. The dataset was composed of 242 lmCRC (172 from center A and 70 from center B) from 57 patients, including: 102 with a baseline long-axis diameter between 10 and <20 mm; 71 and 53 with a long-axis between 20 to 30 mm, and 31 to 50 mm, respectively. The remaining 16 had a long axis of more than 50 mm. Mean lesion diameter was 25 mm (SD: ±14 mm) and no differences in diameter were observed between the training and validation datasets (*p* = 0.09). Conversely, R+ lmCRC showed statistically larger diameters than R− lmCRC (mean diameter of R+ lesions 28 ± 15 mm versus 22 ± 13 mm of R− lesions; *p* < 0.001). Thirty-nine lmCRC had a diameter < 10 mm at TP1 (26 R+ and 13 R−) and were proportionally distributed among the training and validation sets. In the training set, 95 of 172 lesions (55%) were classified as R+, the remaining 77 (45%) as R−; in the validation dataset, 33 of 70 lmCRC (47%) were classified as R+, the remaining 37 (53%) as R−. A heterogeneous or mixed response of lmCRC was observed in 19 of 87 patients (22%), including 15 of the 61 patients (24%) in the training cohort, and 4 out of the 26 patients (15%) in the validation set.

Among the initial cohort of 107 features, 57 were discarded because they were not statistically different between R+ and R− lmCRC according to the Mann–Whitney test, while 27 were discarded since their AUC was not statistically greater than 0.6. Therefore, the FS step retained 23 of the 107 extracted features. When all machine learning techniques were implemented, starting from this features subset, the best performances were reached by the DT, which showed an accuracy of 86% in the validation set versus 63% to 77% of all other ML techniques (more details are presented in Appendix A). The DT is an algorithm that recursively and automatically performs a features selection by using metrics such as information gain to discard noninformative or redundant features during its training. Therefore, the trained DT retained 11 of the 23 previously selected features. The graph of the DT and selected features are shown in Appendix A. Within this algorithm, the best cutoff, i.e., the one that maximizes NPV on the training set, was 0.4. 

The per-lesion performance in predicting response to treatment of our algorithm is shown in Table 1. In brief, a sensitivity and specificity of 99% and 94%, respectively, were obtained in the training set. When the same model and cutoff were applied to the independent validation set at the optimized cutoff, sensitivity and specificity in detecting R+ lesions were 85% and 92%, respectively. The AUC were 0.99 (95% CI = 0.97–1.0) and 0.93 (95% CI = 0.87–0.96) in the training and the validation set, respectively. The accuracy and PPV of the radiomics score on the validation set were not significantly different to the lesion RECIST (accuracy 86% vs. 84%; *p* = 0.68, and PPV 90% vs. 77%; *p* = 0.10, respectively); however, the radiomics score showed higher SP than the lesion RECIST (92% vs. 67%; *p* = 0.003). On the other hand, the lesion RECIST showed higher SE than the radiomics score (100% vs. 85%; *p* < 0.0001).

Figure 2 shows the waterfall plots for all the lmCRC in the training and validation sets using the radiomics score. On both sets, most lesions were predicted either to be R+ with a very high probability (≥98%) or R− with a 100% probability (0% probability to be R+); intermediate probabilities were associated only to a few lesions (19 in the training set and 7 in the validation set). Of note, the classifier is very accurate in separating the two classes, but provides a final classification that is insensitive to the choice of the threshold, which was set at 0.4 but could have ranged between 0.125 and 0.5 without affecting the performances of both training and validation sets.

Table 2 shows lmCRC that were misclassified in the validation dataset. Of the 8 misclassified lesions, three were R− classified as R+ and 5 were R+ classified as R−. The latter included two lmCRC that reached CR after TP1. Figure 3 shows an example of a patient with a mixed response that was correctly classified by our algorithm. Figure 4 shows an example of a patient with all R+ lmCRC in the validation cohort that was wrongly predicted to have two R− lmCRC. CT scans of the remaining misclassified lesions on the validation set (baseline, TP1, and last FU) can be reviewed in Appendix A.

## 4. Discussion

In this study, we developed a delta-radiomics signature based on 11 parameters retained by a DT classifier after subtracting radiomics features at baseline from those at TP1. The signature predicted a long-term response of individual lmCRC with a high diagnostic accuracy, i.e., 97% and 86% in the training and validation dataset, respectively. Importantly, the delta-radiomics signature was able to reliably predict R− lmCRC which were wrongly classified by lesion RECIST as R+ at TP1. Indeed, 106 out of the 114 lesions classified as R− by the signature were confirmed being R− (specificity = 93% averaging training and validation set), whereas if we had based our assessment on lesion RECIST, only 76 of the 114 lesions would have been classified as R− (specificity = 67%). Our results confirm RECIST as a powerful tool to predict disease progression in patients who show lesion-specific PD at TP1 (NPV = 100%), whereas the criteria appear less effective when patients experience early progression (<8 months) after a short period of SD or PR (PPV = 77%). The latter are those that could benefit from delta-radiomics analysis, aimed at anticipating as much as possible a diagnosis of disease progression in order to potentially anticipate alternative treatments. This finding is, in our opinion, of key importance in clinical practice to predict which patients will have a short-term overall response, specifically in cherry picking unresponsive lesions in patients with a heterogeneous response. Liver lesions predicted as resistant at the first follow up in the context of a mixed response could be biopsied and undergo pathological (re)assessment, potentially revealing a different genetic makeup, prompting the use of a different target molecule or of interventional radiology ablative procedures. 

Prior studies have already proposed the use of radiomics signature for per-lesion analysis in different cancer types, including CRC patients [31,32,33]. Giannini et al. [32], for example, validated a radiomics model form baseline CT to predict the behavior of individual lmCRC to targeted treatment in a cohort of HER2 amplified CRC patients. However, results from radiomics analysis performed on baseline CT were sometimes contradictory. For example, in the study by Ahn et al. [17], lmCRC with texture parameters expressing greater structural heterogeneity were associated with worse clinical response to therapy, while Nakanishi et al. [16] and Lubner et al. [18] reported opposite results. 

To the best of our knowledge, this is the first trial to perform per-lesion analysis using a delta-radiomics signature in CRC patients. Of note, two studies validated a delta-radiomics score on lmCRC in order to predict patients OS, and none of them provided external validation [22,23]. Patients’ survival is usually considered the best reference standard in clinical trials, but since it depends also on the patient clinical conditions and subsequent lines of therapy, it cannot be relied on to assess only one line of therapy [34]. Besides, assessment of OS is not applicable in per-lesion analysis and in the case of this study, could be strongly influenced by the presence of extrahepatic disease. We therefore classified liver lesions sensitivity based on a progression-free interval cutoff, which properly fitted our study design.

One of the most critical issues encountered in previous radiomics studies, including the above-mentioned, is the lack of reproducibility [35]. Our study has stacked this issue since it included an external validation cohort. Indeed, our radiomics score was developed using all patients from one center and externally validated using all patients from another center, with the advantage of an extremely reduced risk of overfitting.

This study has several limitations. First, we did not evaluate possible features variations due to different CT equipment and protocols and their impact on features selection. However, since we worked using real-world data, the sample size of each different configuration was not sufficient to compare all differences in acquisition and reconstruction filters. On the other hand, in previous work, we have shown that most radiomics features are robust, being independent of CT parameters [36] and not affected by inter-reader contouring (16). Second, the process of manually segmenting a high number of lesions is time-consuming and still a possible source for errors, since at times liver lesions may be difficult to contour, especially when they show mild hypodensity or ill-defined margins. Indeed, chemotherapy-induced steatosis and shrinking may make lesions almost indistinguishable from surrounding liver parenchyma (see Appendix A). In this study, to validate our radiomics model in a setting mirroring clinical practice, we adopted an intention-to-treat rationale, and therefore chose not to exclude from our analysis “difficult to contour” lesions at TP1. In the future, automatic segmentation with deep learning algorithms will hopefully reduce the burden of segmentation, having also the potential to improve lesion contouring, reduce segmentation time, and improve reproducibility [37,38,39]. Moreover, when larger patient cohorts will be available, deep learning methods, including convolutional/recurrent/artificial neural networks, could be exploited to predict response to therapy as an alternative to machine learning algorithms [40]. Third, this study enrolled metastatic CRC patients with a wide range of background medical conditions, with different performance status and comorbidities. Nevertheless, our signature achieved high accuracy in both datasets, proving to be valid even when different therapeutic protocols based on FOLFOX are used. To increase its clinical value, we are currently preparing to test this radiomics score in patients treated with other drugs, including FOLFIRI, the second conventional regimen in mCRC, or undergoing other lines of therapy. Fourth, in this cohort, patients with extrahepatic disease were included. The interruption of first-line chemotherapy due to extra-hepatic progression of the disease, together with high-grade toxicity or death, led to the exclusion of almost 40% of metastases from the initial cohort of patients, including good responder lesions that did not reach 8 months of first-line chemotherapy. Those lesions could not be considered as R+ safely because, as we demonstrated, lmCRC often progress despite their initial response. A cohort of CRC patients with liver-only disease would ideally provide the best sample for our research. Finally, R− lesions were not biopsied; therefore, in this cohort it will not be possible to determine whether differences in prognosis are related to specific molecular drivers. Future studies should combine radiomics features, pathology, and molecular findings into one model. In perspective, an interventional trial could be envisaged, where biopsy is performed in patients with mixed response, to assess the molecular makeup of outlier progressing lesions, triggering adjustments of their therapeutical regimen.

## 5. Conclusions

In conclusion, in this study, we have shown that delta-radiomics is very accurate in predicting the response of lmCRC to first-line FOLFOX chemotherapy in CRC patients. Our approach could pinpoint lesions with distinct biological and molecular features. As such delta-radiomics enables further studies toward lesion-specific personalized treatment in liver-only metastatic colorectal cancer. Further validation of our radiomics biomarker on different cohorts of patients is warranted.

## Figures and Tables

**Figure 1 cancers-14-00241-f001:**
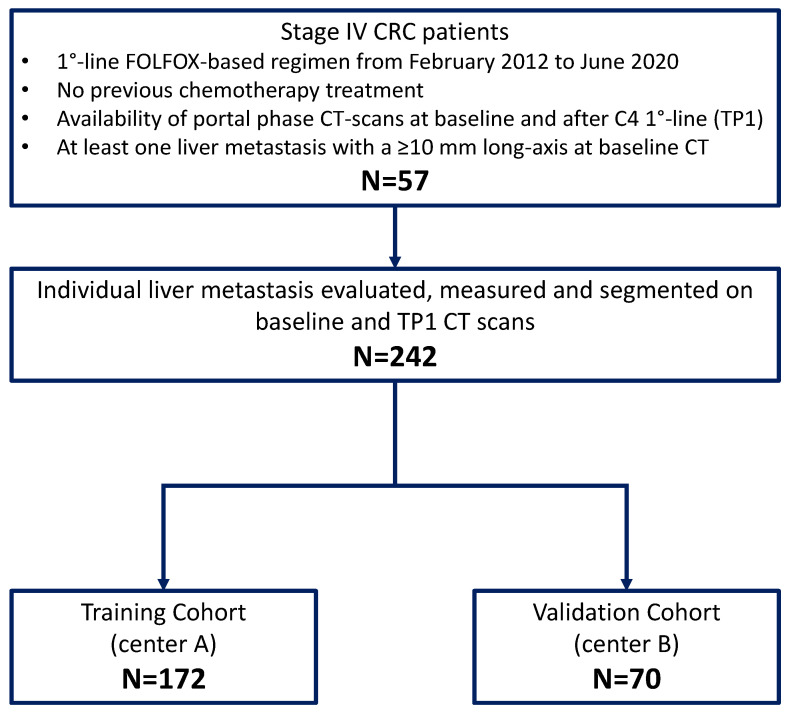
CONSORT flow diagram.

**Figure 2 cancers-14-00241-f002:**
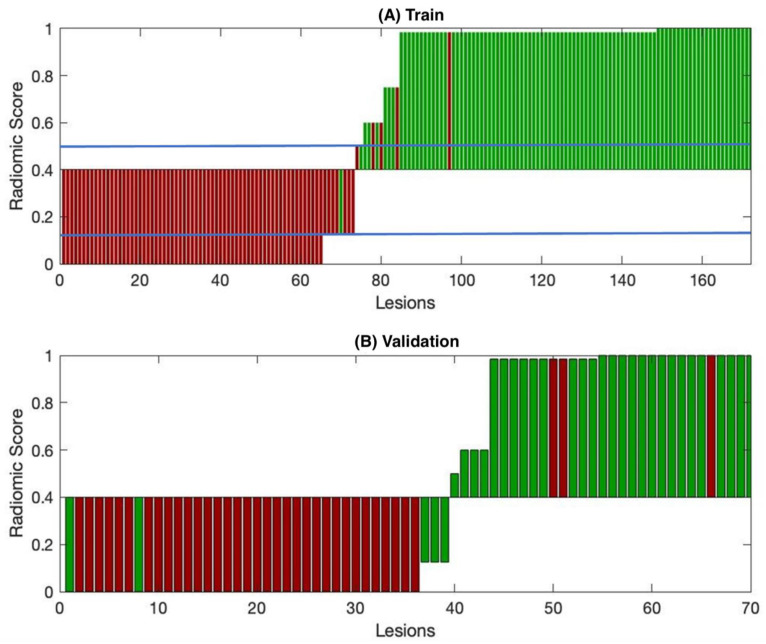
Waterfall plot of lesions included in the training (**A**) and validation sets (**B**). The green marks indicate the responder lesions, while the red marks represent the nonresponder lesions. All errors in the validation set (8) are visible as miscolored bars. The *y*-axis represents the radiomics score produced by the decision tree. The radiomics optimized cutoff is 0.4. Blue lines show the range (0.125–0.5) in which the cutoff could be modified without changing the prediction.

**Figure 3 cancers-14-00241-f003:**
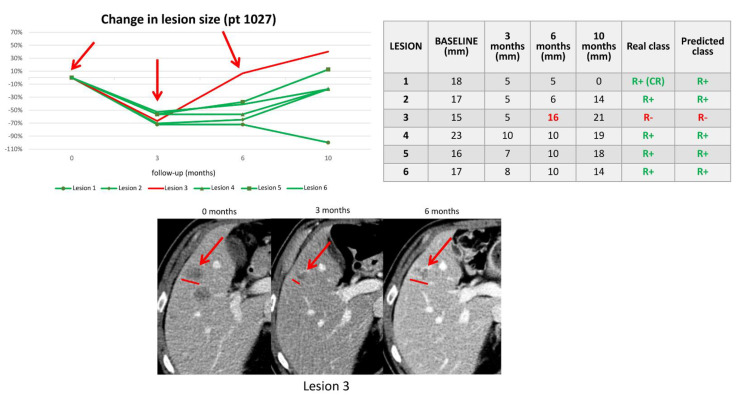
In patient 1027, our algorithm correctly classified a R+ 5 liver metastases that responded to therapy for 10 months and classified a R− 1 liver lesion (i.e., lesion 3) that showed a PD after 6 months of treatment. The red line in the graph represents the metastasis that went in progression before 8 months, while green lines represent metastases that respond for at least 8 months. The table lists the patient’s liver metastases, size at baseline, and subsequent timepoints. The real and predicted class columns show the 8-month response of each lesion based on size variations and the classification as predicted by the classifiers.

**Figure 4 cancers-14-00241-f004:**
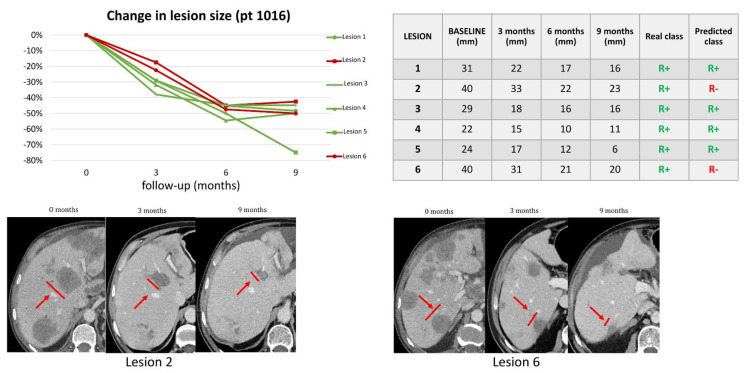
In patient 1016, our algorithm correctly classified a R+ 4/6 lesions while 2/6 lesions were misclassified as R−. The red line in the graph represents the metastasis that went into progression before 8 months, while the green lines represent metastases that responded for at least 8 months The table lists the patient’s liver metastases, size at baseline, and subsequent timepoints. The real and predicted class columns show respectively the 8-month response of each lesion based on size variations and the classification as predicted by the classifiers.

**Table 1 cancers-14-00241-t001:** Decision tree performances on training and validation set.

	**ACC %**	**SE %**	**SP %**	**PPV %**	**NPV %**
	**(95% CI)**	**(95% CI)**	**(95% CI)**	**(95% CI)**	**(95% CI)**
	**[Rate]**	**[Rate]**	**[Rate]**	**[Rate]**	**[Rate]**
Train	97	99	94	95	99
	(89–100)	(94–99)	(85–98)	(89–98)	(91–100)
	[166/172]	[94/95]	[72/77]	[94/99]	[72/73]
Validation	86	85	92	90	87
	(81–92)	(68–95)	(78–98)	(76–96)	(75–94)
	[62/70]	[28/33]	[34/37]	[28/31]	[34/39]
Lesion RECIST	84	100	67	77	100
	(79–87)	(97–100)	(57–75)	(72–81)	(79–88)
	[202/242]	[128/128]	[76/114]	[128/166]	[76/76]

**Table 2 cancers-14-00241-t002:** Misclassified lesions of the validation set. Time of follow up (FU) is reported when the lesion is not in complete response.

PZ	LES	REAL CLASS	Baseline Diameter (mm)	TP1 Diameter (mm)	Last FU Diameter (mm)	Time of FU
1010	7	R−	33	10	33	6 months
	10	R−	33	13	33	6 months
1017	2	R−	44	28	32	6 months
	4	R+	14	13	0	CR
1001	1	R+	11	6	0	CR
1015	4	R+	28	18	12	9 months
1016	2	R+	40	33	23	9 months
	6	R+	40	31	20	9 months

## Data Availability

The data presented in this study are available on request from the corresponding author. The data are not publicly available due to privacy reasons.

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
