# Peer review of "Delta-Radiomics Predicts Response to First-Line Oxaliplatin-Based Chemotherapy in Colorectal Cancer Patients with Liver Metastases"

_cancers, 2022, doi:10.3390/cancers14010241_

Round 1

Reviewer 1 Report

This is an outstanding effort with promising results. However, radiomics is a complex concept for clinicians, and therefore I suggest adding some language in the methodology that can be more easily interpreted. More importantly, this plainer language should help the reader understand the development process from more than 100 variables tested to a final score of 11, and how the optimization of these 11 variable cut-off points was performed. Lastly, it is critical to clarify that the score involved the variables and the cut-off points for each variable (or how the score is calculated, do variable individual scores weigh the same or not, etc) and that these were not changed in the independent validation set, even when these patients were treated at a different institution and had undergone different CT scans, it is important to clarify that no interplatform retraining was performed, or if so, acknowledge this as a retaining. In other words, is this delta signature with its current cut-off points of each variable (or is it an asymmetric score?) applicable to any other patient with any other CT scan hardware and software. 

Author Response

COMMENT 1: This is an outstanding effort with promising results. However, radiomics is a complex concept for clinicians, and therefore I suggest adding some language in the methodology that can be more easily interpreted. More importantly, this plainer language should help the reader understand the development process from more than 100 variables tested to a final score of 11, and how the optimization of these 11 variable cut-off points was performed.

RESPONSE 1: We thank the reviewer for the useful comments. We better explained the features selection steps in both the M&M (section2.5) and results (Second paragraph). In particular, we reported the number of features that were discarded since the Mann-Whitney test did not consider them as statistically different between R+ and R- lmCRC, and the number of features that were discarded since their AUC was lower than 0.6. We also better explained why 0.6 has been chosen, i.e., to consider only features that individually perform statistically better than a random classifier, i.e., AUC=0.5. Finally, we reformulated the sentence in which we described that the Decision tree algorithm automatically performs a features selection by using metrics such as information gain, to discard non-informative or redundant features, thus retaining 11 out of 23 previously selected features (second paragraph of the results section).

COMMENT 2: Lastly, it is critical to clarify that the score involved the variables and the cut-off points for each variable (or how the score is calculated, do variable individual scores weigh the same or not, etc) and that these were not changed in the independent validation set, even when these patients were treated at a different institution and had undergone different CT scans, it is important to clarify that no interplatform retraining was performed, or if so, acknowledge this as a retaining. In other words, is this delta signature with its current cut-off points of each variable (or is it an asymmetric score?) applicable to any other patient with any other CT scan hardware and software. 

RESPONSE 2: We thank the reviewer, and we confirm that all parameters, cut-offs and weights were trained and selected during the training step and used as they were on the validation sets. Therefore, this delta-radiomics score can be applied, without any tuning, to any other patient with any other CT scan hardware and software. Thanks to this comment, we better explained that in section 2.5.

Reviewer 2 Report

Groundbreaking work regarding the use of AI/Delta-radiomics to predict response to oxaliplatin based chemoterapy as a first line therapy in lmCRC. Indeed, this is an innovative study with very promising results - per-lesion sensitivity, specificity, PPV and NPV were 99%, 94%, 95%, 99% and 85%, 92%, 90%, 87% respectively in the training and validation datasets. The further application and the clinical evaluation of the usefulness of the AI tool developed by this group should be considered in a future multicentric study. Congratulations!

Author Response

COMMENT: Groundbreaking work regarding the use of AI/Delta-radiomics to predict response to oxaliplatin based chemoterapy as a first line therapy in lmCRC. Indeed, this is an innovative study with very promising results - per-lesion sensitivity, specificity, PPV and NPV were 99%, 94%, 95%, 99% and 85%, 92%, 90%, 87% respectively in the training and validation datasets. The further application and the clinical evaluation of the usefulness of the AI tool developed by this group should be considered in a future multicentric study. Congratulations!

RESPONSE: We thank the reviewer for carefully reading our work and for acknowledging the promising results we obtained. We are already planning to validate this in a multicentric study. 

Reviewer 3 Report

Acceptance after minor revisions

In my opinion the present manuscript, entitled “Delta-Radiomics Predicts Response to First-line Oxaliplatin-Based Chemotherapy in Colorectal Cancer Patients with Liver Metastases” will be a great contribution to the scientific community. The combination of radiomics with machine learning represents a promising strategy for the area of precision oncology. Giannini et al. have successfully validated a per-lesion approach to predict non-responders and good responders among lmCRC patients towards FOLFOX regimen. Please, find below some comments to improve the publication:

1) In the abstract, the abbreviation TP1 must be explained since its meaning appears only in the sixth paragraph of the introduction section. There are other abbreviated forms in the abstract section without explanation (e.g., CT and FOLFOX), but in my view they might be left like that.

2) Section 2.1 / Line 3 >>> Typo “…data were retreated…” (not where)

3) Section 2.2 / Line 3 >>> Please, indicate which types of contrast medium were used. Was this another limitation of the study? If so, please, make it clearer in the paper.

4) Section 2.4 / First paragraph / Lines 5-7 >>> How many times the segmentation made by the residents had to be, in fact, modified by the experienced radiologists? Please, clarify this information in the result section.

5) Section 2.5 / Second paragraph / Line 2 >>> Typo (words without space)

6) Results / First paragraph / Lines 6 and 7 >>> Please, reformulate the sentence. Which parameter of lmCRC is being explained? Even though the reader gets the idea that is diameter, the way of writing needs to be corrected.

7) Results / Second paragraph / Lines 1-3 >>> How was the accuracy of the other 3 machine learning techniques? Please, indicate how was the performance of them as well, especially emphasizing how better DT performed to be selected. The accuracy of all 4 techniques could be placed together in a supplementary table.

8) Results / Fifth paragraph / Line 1 >>> Typo (correct is “misclassified”)

9) Reference 13 >>> Citing the original paper would be the best approach, instead of the erratum version.

10) Discussion >>> The use of deep learning was mentioned in the discussion for the improvement of segmentation (References 37-39). However, this technique (e.g., convolutional, recurrent, artificial neural network, etc) must be also discussed as a possible alternative to the decision tree algorithm selected in the paper.

Author Response

COMMENT: In my opinion the present manuscript, entitled “Delta-Radiomics Predicts Response to First-line Oxaliplatin-Based Chemotherapy in Colorectal Cancer Patients with Liver Metastases” will be a great contribution to the scientific community. The combination of radiomics with machine learning represents a promising strategy for the area of precision oncology. Giannini et al. have successfully validated a per-lesion approach to predict non-responders and good responders among lmCRC patients towards FOLFOX regimen. Please, find below some comments to improve the publication:

MULTI-RESPONSE: We thank the reviewer, and we provide a point-by-point response.

1) In the abstract, the abbreviation TP1 must be explained since its meaning appears only in the sixth paragraph of the introduction section. There are other abbreviated forms in the abstract section without explanation (e.g., CT and FOLFOX), but in my view they might be left like that.

Thank you, we corrected it.

2) Section 2.1 / Line 3 >>> Typo “…data were retreated…” (not where)

Thank you, we corrected it.

3) Section 2.2 / Line 3 >>> Please, indicate which types of contrast medium were used. Was this another limitation of the study? If so, please, make it clearer in the paper.

We thank the reviewer for this comment. Unfortunately, since the dataset is composed of real-word cases, different contrast medium and with different composition in term of mg/ml (mg of Iodine over mL) were used. However, the dose was selected based on patient’s weight to obtain the same quantity of Iodine. That said, in our opinion this is not a limitation since performance of the classifier was not influenced by contrast agent molecule and concentration, nor by differences in protocols and scanners. Thank to this comment, we added more information on contrast medium in section 2.2

4) Section 2.4 / First paragraph / Lines 5-7 >>> How many times the segmentation made by the residents had to be, in fact, modified by the experienced radiologists? Please, clarify this information in the result section.

We thank the reviewer for this suggestion. We added the percentages of segmentations that were revised by an experienced radiologist. In particular, we reported in section 2.4 that approximately 50% of baseline segmentations were slightly revised by an experienced radiologist for millimeter modifications, while 5% of them were strongly revised. Conversely, at TP1 only 10% of segmentations were slightly revised. Therefore, we believe that an experienced radiologist should revise the segmentations but also that the learning curves are quite steep.

5) Section 2.5 / Second paragraph / Line 2 >>> Typo (words without space)

Thank you, we corrected it.

6) Results / First paragraph / Lines 6 and 7 >>> Please, reformulate the sentence. Which parameter of lmCRC is being explained? Even though the reader gets the idea that is diameter, the way of writing needs to be corrected.

Thank you, we reformulated the sentence better explaining that the variable was the diameter.

7) Results / Second paragraph / Lines 1-3 >>> How was the accuracy of the other 3 machine learning techniques? Please, indicate how was the performance of them as well, especially emphasizing how better DT performed to be selected. The accuracy of all 4 techniques could be placed together in a supplementary table.

Thank you for this useful comment. We revised the text and added a supplementary table with the other ML performances.

8) Results / Fifth paragraph / Line 1 >>> Typo (correct is “misclassified”)

Thank you, we corrected it.

9) Reference 13 >>> Citing the original paper would be the best approach, instead of the erratum version.

Thank you, we corrected it.

10) Discussion >>> The use of deep learning was mentioned in the discussion for the improvement of segmentation (References 37-39). However, this technique (e.g., convolutional, recurrent, artificial neural network, etc) must be also discussed as a possible alternative to the decision tree algorithm selected in the paper.

The reviewer is right. We add a sentence and a reference in the discussion.